# Preliminary Study on Early Diagnosis and Rehabilitation Treatment of Pine Wood Nematode Disease Based on Partial Symptoms

**Anshun Ni, Dan Yang, Hao Cheng and Jianren Ye ***

Co-Innovation Center for Sustainable Forestry in Southern China, College of Forestry,
Nanjing Forestry University, Nanjing 210037, China
* Correspondence: jrey@njfu.edu.cn

**Abstract:** Pine wilt disease (PWD) is greatly serious to *Pinus*, and there are still no effective therapeutic measures at present. It is necessary to explore a method of early diagnosis of PWD and to study rehabilitation treatment technology for diseased plants diagnosed early. This paper searched for infected pine trees in natural pine forests according to various subtle symptoms and divided the disease development stages. Different doses of 20% emamectin benzoate were injected at different stages, and the symptom development of pine trees was observed after injection. According to different external symptoms, the stage after being infected by PWD was divided into early stage I and II, middle stage I and II, late stage. It was shown by the results that the diagnostic rate of initial diagnosis based on the symptoms of early stage II was as high as 80%. Additionally, for early stage infected pine trees, an injection of 5–10 mL of 20% emamectin benzoate can inhibit the expansion of symptoms of PWD. One year after injection of 20 mL and 10 mL of 20% emamectin benzoate, the residues measured in the lateral branches 4 m above the injection point were 0.18 mg kg$^{-1}$ and 0.06 mg kg$^{-1}$, respectively. In summary, the characteristics of early stage II are ideal for identifying early infection, it has a certain therapeutic effect on early infected pine trees by injection of emamectin benzoate.

**Keywords:** pine wood nematode disease; external symptoms; early diagnosis; trunk injection; treatment of diseased tree





## 1. Introduction

Pine wood nematode disease, also known as pine wilt disease (PWD), is caused by the plant–parasitic nematode *Bursaphelenchus xylophilus* (Nickle) and is a systemic disease with a wide host range, rapid spread, and rapid death after infection [1–4]. The disease originated in North America and was first discovered in Japan in 1905 [5] and later spread from Japan to China, Korea, and other regions. In China, PWD was first reported in Nanjing in 1982 [6], and since then, it has spread rapidly throughout the country, causing huge economic losses [7,8].

The current prevention and control measures of PWD mainly include epidemic monitoring, disease quarantine, vector beetle control, removal of diseased wood, and trunk injection [9], which are all preventive measures against PWD [10–12], and lack of rehabilitation and treatment measures for pine trees after being infected. Treatment techniques are important for controlling PWD, especially for the protection of ancient and famous pine trees. The main difficulty in the rehabilitation of PWD is the early diagnosis of the infected trees and screening of therapeutic agents that can be used in the growing season [13].

The key to early diagnosis of PWD lies in the ability to accurately diagnose PWD before the sap in the trunk of infected pine trees stops flowing; theoretically, the earlier the disease is diagnosed, the more likely it is to be cured. At present, there is a lack of effective early diagnostic techniques for pine trees after infection with *B. xylophilus* both at home and abroad [14–17]. For the therapeutic agents for infected trees, because PWD starts to spread

in the tree growth season, the chemical agents used for the treatment of diseased trees must first be absorbed by the tree in the tree growth season [18]. At present, most commercially available pine tree trunk injection agents are only used in nongrowing seasons.

The aim of this study was to find a method for the early diagnosis of PWD by comparing the early manifestations of subtle external symptoms of infected pine trees and, based on this information, to find trunk injection agents that can still be absorbed by the tree during the growing season for therapeutic testing. Additionally, we aimed to explore the possibility of early diagnosis and rehabilitation treatment at different stages of infection to provide possible technical methods for the treatment of pine trees infected by *B. xylophilus*.

## 2. Materials and Methods

### 2.1. Experimental Sample Sites and Materials

The experimental site was selected in the 19-year-old *Pinus densiflora* and *Pinus thunbergii* experimental stands in the Yaolingkou work area (119°12′ E, 32°05′ N) of Jurong tree Farm, Jiangsu Province. The trunk injection agent used in the test was 20% emamectin benzoate SL, which can be injected into the tree body in the tree growing season based on initial testing performed by our team. The average diameter at breast height (DBH) of the treated pine trees was 16 cm, and the average tree height was 7.5 m.

### 2.2. Selection of Diseased Pine Trees

From May 2021 to September 2022, suspected infected branches were collected in the experimental stands based on signs of the feeding wounds of longicorn beetles, discolouration of needles, oviposition scar of longicorn beetles, and the condition of resin flux. The branches were taken back to the laboratory, and the nematodes were isolated with the Baermann funnel method. Morphological identification of the nematodes was conducted under a microscope, and detection with a PWN molecular detector was used to determine the presence of *B. xylophilus*.

### 2.3. Morphological Identification of B. xylophilus

The PWN suspension obtained by the Baermann funnel method and centrifuged 3500 r/min for 3 min by centrifuge 5804 R (Eppendorf), and the precipitate was washed with sterile water for 2 times. Then, the PWN suspension at the bottom of the 15 mL centrifuge tube was removed and then observed under an optical microscope. Nematode morphology was identified by observing the nematode head, tail, mouth needle, oesophageal gland, and reproductive system [19,20].

### 2.4. Molecular Detection of B. xylophilus

After observing the suspected PWN under a light microscope, 20 μL was transferred to a 1.5 mL centrifuge tube, 20 μL of nematode DNA extract solution A and 2 μL of nematode DNA extract solution B were added, shaken, and mixed well; the centrifuge tube was placed into a metal bath, heated at 95 °C for 45 min and then at 65 °C for 10 min, and centrifuged at 12,000 r/min for 3 min. Then, 3.5 μL of supernatant was aspirated into a 200 μL PCR tube, 5 μL of pine wood nematode detection reagent K solution and 1.5 μL of pine wood nematode detection reagent P solution were added, mixed well, centrifuged briefly, and placed in a Bx-48 PWN automated molecular detector (Hangzhou Bioer Technology Co., Ltd., Hangzhou, China) for molecular detection [21].

### 2.5. Medication Treatment

Medication treatment for pine trees confirmed to be infected. In 2021, gravity flow trunk injection was adopted for treatment. The steps were as follows: drilling a hole 10 cm deep at an angle of 45 degrees at the breast height of the diseased pine tree, inserting the injection bottle, piercing the hole at the bottom of the bottle, and observing the absorption of the agent after one week. In 2022, the direct injection method was adopted for treatment, with the following steps: drilling 10 cm deep with an 8 mm drill bit at an angle of 45° at the

breast height of the pine tree, directly injecting agents into the drilling hole with a liquid transfer gun and plugging the drilling hole with plasticine.

*2.6. Accuracy of Initial Diagnosis*

According to the external symptoms of infected pine trees, samples of suspected infected trees were collected, confirmed by nematode morphology and molecular detection, and the accuracy of initial diagnosis was obtained according to the ratio of suspected infected trees and infected trees. The calculation formula is:

Accuracy of initial diagnosis = (infected trees/suspected infected trees) × 100%.

*2.7. Detection of Pesticide Residues*

One year after trunk injection, the diseased pine trees were sampled from the lateral branches 4 m above the injection point, and the pesticide residues in the trees were detected by an Agilent 1290–6470 liquid chromatography-mass spectrometer.

*2.8. Data Processing*

Divide the drug therapeutic effect into two groups and use chi-square test to compare and describe. $p < 0.05$ indicates that the difference is statistically significant.

**3. Results**

*3.1. Observation of External Symptoms in the Early Stage of Infection and Division of the Stage of Infection*

The external abnormal symptoms related to PWD include feeding wounds of vector beetles on twigs, needle discolouration on local twigs, and oviposition scars of vector beetles on the trunk (Figure 1). The feeding wounds of beetles usually first appear on the twigs of pine trees. After the bite, the PWN carried by the vector beetle can invade the twigs from the wound. After a period of time, the needles on some twigs begin to change colour, and the nematodes multiply in the twigs and gradually turn into the trunk, leading to decreases in the whole pine resin secretion, weakening the tree vigour, attracting the vector beetles to lay eggs, and leaving a groove for longicorn to lay eggs at the trunk of the weakened pine tree. Finally, the whole pine tree withers.

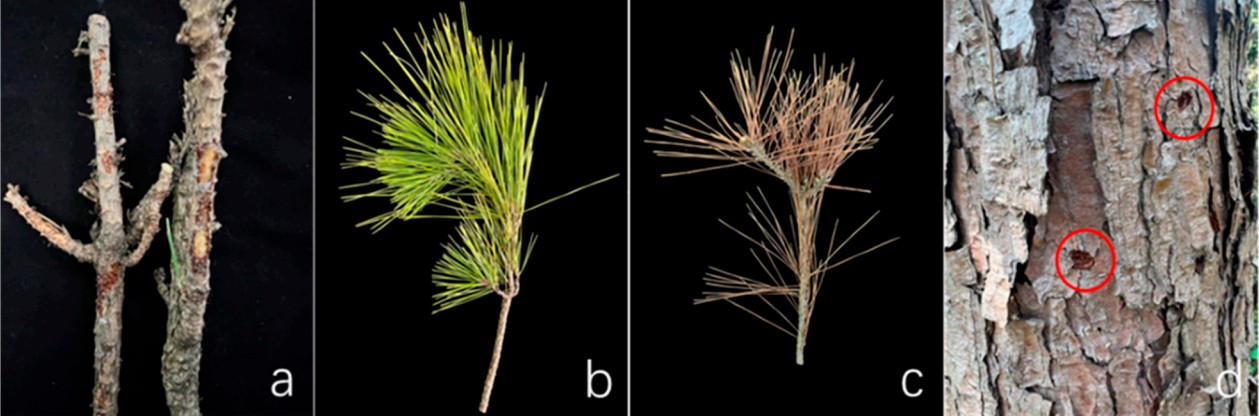

**Figure 1.** External symptoms of pine trees infected with *B. xylophilus*. (**a**). Beetle feeding wound; (**b**). Needle discolouration; (**c**). Needle wilting; (**d**). Red circle indicates vector beetle oviposition scars.

The infected pine trees were classified into 5 categories based on the external symptoms of PWD they exhibited (Table 1): (1) in early stage I, there were feeding wounds of vector beetles, but the needles did not change colour; (2) in early stage II, the needles had slight discolouration on a single branch; (3) in middle stage I, only a single twig wilted; (4) in middle stage II, needle discolouration and withering was evident on several branches, and

the withering rate of the whole plant was less than 50%, and (5) in late stage, more than 50% of the needles on the branches wilted or there were a large number of vector beetle oviposition scars on the trunk. We concluded that pine trees with no obvious discolouration of needles may still be in the late stage of disease.

**Table 1.** Division of disease stages of pine wilt disease. "+" indicates the degree of pine resin flow, the more "+" means more pine resin, "-" means no pine resin.

| Stages of Disease | External Symptoms | Condition of Resin |
|---|---|---|
| Early stage I | maturation feeding evident on twigs, no needle discolouration | ++++ |
| Early stage II | maturation feeding evident on twigs, Slight discolouration of needles | +++ |
| Middle stage I | maturation feeding evident on twigs, Single twig wilting | + |
| Middle stage II | Needle wilt less than 50%, a few oviposition scars | - |
| Late stage | Needle wilt greater than 50% or whole plant discolouration or many Monochamus oviposition scars | - |

The study found that in early stage I of the PWD, the pine resin in the pine tree was normal; the amount of pine resin in early stage II was slightly reduced; the pine resin in middle stage I was significantly reduced, and in middle stage II, no pine resin was detected. In the late stage of the disease, the xylem of the tree had a blue stain.

### 3.2. The Diagnostic Accuracy at Different Stages of PWD

The suspected and confirmed diseased trees in the two-year trial in 2021 and 2022 are shown in Table 2. Thirty-five suspected infected trees were found in 2021, of which 13 were confirmed to have the disease. In 2022, 36 suspected infected trees were found, of which 16 were confirmed to have the disease. Based on the results of the two years, the diagnostic accuracy at each stage is shown in Table 3. The table shows that the total diagnosis accuracy of pine trees with early stage I and II symptoms was 25.9%. The diagnosis accuracy of early stage I was only 20.4%, while the diagnosis accuracy of early stage II was as high as 80%, which was significantly higher than that of early stage I. The diagnosis accuracy was 84.6% when multiple branches were discoloured (discolouration percentage < 50%) in the middle stage of the disease. The diagnosis accuracy reached 100% when the percentage of discolouration of branches was >50% in the late stage of the disease and there were oviposition scars of longicorn beetles. In addition, a large number of pine wood nematodes were observed under the optical microscope when sampling from the side branches and trunk of the four pine trees in the late stage of disease.

**Table 2.** Suspected and confirmed infected pine trees in 2021 and 2022.

| Stages of Disease | Suspected Infected Trees in 21 Years | Infected Trees in 21 Years | Suspected Infected Trees in 22 Years | Infected Trees in 22 Years |
|---|---|---|---|---|
| Early stage I | 24 | 4 | 25 | 6 |
| Early stage II | 3 | 2 | 2 | 2 |
| Middle stage I | 1 | 1 | 1 | 1 |
| Middle stage II | 7 | 6 | 4 | 3 |
| Late stage | 0 | 0 | 4 | 4 |
| Total | 35 | 13 | 36 | 16 |

**Table 3.** Total accuracy of the initial diagnosis of each disease stage in 2021 and 2022.

| Stages of Disease | Suspected Infected Trees | Confirmed Infected Trees | Accuracy of Initial Diagnosis (%) |
|---|---|---|---|
| Early stage I | 49 | 10 | 20.4 |
| Early stage II | 5 | 4 | 80.0 |
| Middle stage | 13 | 11 | 84.6 |
| Late stage | 4 | 4 | 100.0 |

*3.3. The Effect of Pharmaceutical Treatment at Different Stages of PWD*

A total of 13 suspected diseased pine trees were confirmed to have different stages of disease in 2021, including 6 early stage and 7 middle stage pine trees (Table 4). One early stage II and two middle stage II pines were used as the control group without injection. The remaining 5 plants in the early stage (4 plants in early stage I and 1 plant in early stage II) were injected with 10 mL 20% emamectin benzoate SL (Figure 2A–E). One pine tree was injected with 10 mL at middle stage I (Figure 2F), and 2 of the 4 pine trees at middle stage II were injected with 20 mL (Figure 2G,H), and the other 2 were injected with 10 mL (Figure 2I,J). The incidence of pine trees was observed and recorded every other month. The results showed that the 3 trees in the control group all died within 3 months. The five pine trees injected with 20% emamectin benzoate (10 mL) at the early stage of the disease did not die, and their symptoms had not developed further as of 29 September 2022 (Figure 2a–e). One middle stage I pine tree injected with 10 mL (Figure 2f) did not die in that year, and by 29 September 2022, the symptoms had progressed to those of late stage disease. Two middle stage II pine trees injected with 10 mL 20% emamectin benzoate both died in 2021, and two meddle stage II pine trees injected with 20 mL 20% emamectin benzoate both were still alive in 2021 until the next year, 13 September 2022, when one of them died, and the other tree was not only alive until 29 September 2022 but also had no obvious signs of symptom progression (Figure 2g,h).

**Table 4.** The dosage of trunk injection for different confirmed infected pine trees in 2021 and 2022.

| Injection Dose/mL | July August 2021 | | July August 2022 | | |
|---|---|---|---|---|---|
| | Early Stage | Middle Stage | Early Stage | Middle Stage | Late Stage |
| 20 | 0 | 2 | 0 | 0 | 4 |
| 15 | 0 | 0 | 0 | 4 | 0 |
| 10 | 5 | 3 | 0 | 0 | 0 |
| 5 | 0 | 0 | 7 | 0 | 0 |
| 0 (CK) | 1 | 2 | 1 | 0 | 0 |

A total of 16 suspected diseased pine trees were confirmed to have the disease in 2022, including 8 early-stage, 4 middle stage and 4 late stage trees (Table 4). An early stage I pine tree was selected as a control without injection, resulting in the whole tree withering and dying on 29 September of that year (Figure 3h). The other 7 early stage pines (5 early stage I and 2 early stage II) were all injected with 5 mL of 20% emamectin benzoate, resulting in no further symptom development on 29 September 2022 (Figure 3). Four middle stage diseased pines were injected with 15 mL of 20% emamectin benzoate, resulting in the death of two middle stage II pines and no further development of symptoms in one middle stage II and one middle stage I pine by 29 September 2022. Four late stage pine trees were injected with 20 mL of 20% emamectin benzoate and still died within 1 month after injection.

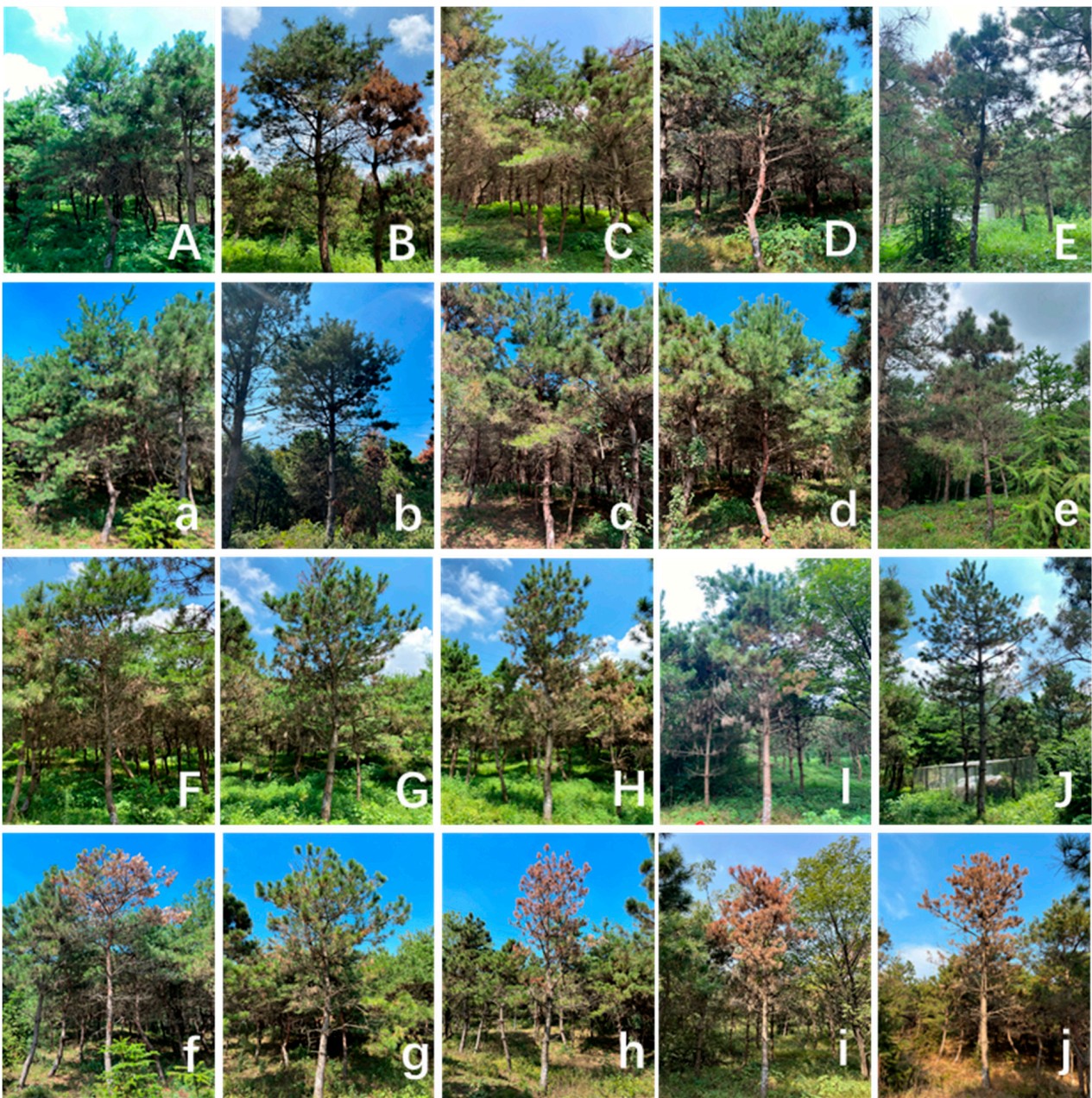

**Figure 2.** Effect of trunk injection treatment of infected pine trees in 2021. a-1 represents the time of shooting (year-month-day): (**A**). Injection 10 mL, 30 July 2021; (**B**). Injection 10 mL, 3 August 2021; (**C**). Injection 10 mL, 6 August 2021; (**D**). Injection 10 mL, 10 August 2021; (**E**). Injection 10 mL, 13 July 2021; (**F**). Injection 10 mL, 3 August 2021; (**G**). Injection 20 mL, 13 July 2021; (**H**). Injection 20 mL, 13 July 2021; (**I**). Injection 10 mL, 10 August 2021; (**J**). Injection 10 mL, 3 August 2021; (**a**). Same tree as (**A**), 29 September 2022; (**b**). Same tree as (**B**), 29 September 2022; (**c**). Same tree as (**C**), 29 September 2022; (**d**). Same tree as (**D**), 29 September 2022; (**e**). Same tree as (**E**), 29 September 2022; (**f**). Same tree as (**F**), 29 September 2022; (**g**). Same tree as (**G**), 29 September 2022; (**h**). Same tree as (**H**), 13 September 2022; (**i**). Same tree as (**I**), 25 September 2021; (**j**). Same tree as (**J**), 23 October 2021. (**A**–**D**) are early stage I pine trees, (**E**) is early stage II pine trees, (**F**) is middle stage I pine trees, (**G**–**J**) are middle stage II pine trees.

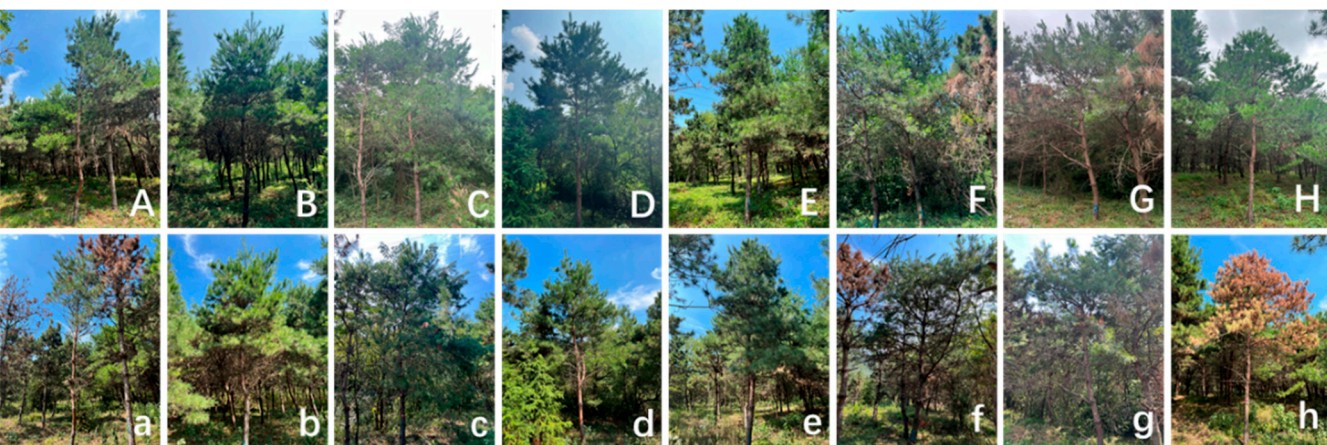

**Figure 3.** Effect of trunk injection treatment of infected pine trees in 2022. a-1 represents the time of shooting (year-month-day): (**A**). Injection 5 mL, 18 July 2022; (**B**). Injection 5 mL, 11 July 2022; (**C**). Injection 5 mL, 18 July 2022; (**D**). Injection 5 mL, 2 August 2022; (**E**). Injection 5 mL, 11 July 2022; (**F**). Injection 5 mL, 11 July 2022; (**G**). Injection 5 mL, 3 July 2022; (**H**). CK, no injection, 18 July 2022; (**a**). Same tree as (**A**), 29 September 2022; (**b**). Same tree as (**B**), 29 September 2022; (**c**). Same tree as (**C**), 29 September 2022; (**d**). Same tree as (**D**), 29 September 2022; (**e**). Same tree as (**E**), 29 September 2022; (**f**). Same tree as (**F**), 29 September 2022; (**g**). Same tree as (**G**), 29 September 2022; (**h**). Same tree as (**H**), 29 September 2022. (**A–E,H**) are early stage I pine trees, (**F,G**) are early stage II pine trees.

Tests have shown that for late stage infected pine trees, the injection agent no longer has a significant therapeutic effect. For pine trees in the middle stage of the disease, injecting high doses of emamamectin benzoate (20 mL 20% emamectin benzoate) may still inhibit the progression of the disease and have a certain therapeutic effect. For early-stage diseased pine trees, the effect of emamectin benzoate is obvious, and injecting 5 mL 20% emamectin benzoate can inhibit the disease and cause the infected pine trees to recover. Chi-square analysis (Table 5) showed that drug injection at different stages of infection was significantly related to the therapeutic effect.

**Table 5.** Analysis of therapeutic effect at different stages of infection.

| Stage of Infection | Therapeutic Effect | | $\chi^2$ | *p* Value |
| --- | --- | --- | --- | --- |
| | **Effective** | **Ineffective** | | |
| Early stage | 12 | 0 | | |
| Middle stage | 1 | 8 | 21.439 | $p < 0.001$ |
| Late stage | 0 | 4 | | |

*3.4. Residual Analysis of Drug in Pine Trees One Year after Injection of Emamectin Benzoate into Diseased Pine Trees in 2021*

The results of the residue testing in pine trees that remained healthy after one year of injection treatment in 2021 are as follows: one year after the injection of 20 mL and 10 mL of 20% emamectin benzoate, the residues measured at the lateral branches 4 m above the injection point were 0.18 mg kg$^{-1}$ and 0.06 mg kg$^{-1}$, respectively (Table 6). This showed that the drug can be transported to the top branches in the tree body after injection in the growing season, and there is still a certain amount of residual after one year. This residual concentration is undoubtedly of positive significance for the continuous recovery of infected pine trees and the prevention of pine wood nematode reinfection.

**Table 6.** Residue of 20% emamectin benzoate in the tree after one year of injection.

| Injection Dose (mL) | Pesticide Residue (mg/kg) | | | Mean (mg/kg) | Standard Deviation | Coefficient of Variation/% |
|---|---|---|---|---|---|---|
| 20 | 0.181 | 0.176 | 0.183 | 0.180 | 0.004 | 2.175 |
| 10 | 0.060 | 0.058 | 0.062 | 0.060 | 0.002 | 3.975 |

## 4. Discussion

This study showed that early diagnosis within a therapeutic window can be obtained by observing local symptomatic manifestations of pine tree infection with PWD for initial diagnosis, followed by morphological or molecular tests to confirm the diagnosis. The initial diagnosis is based on the external symptoms of the first infected twigs with feeding wounds of longicorn beetles and slight discolouration of the needles, and the accuracy rate was more than 80%. For the early stage of infected pine trees, timely injection treatment (such as 20% emamectin benzoate, 5–10 mL, general dose) can not only stop the further development of the disease, but also return the infected pine trees to health. The diagnosis rate can reach more than 84% according to the initial diagnosis based on the bite wound of longicorn beetles on the infected pine tree and the symptom of multiple branch discolouration (discolouration percentage < 50%). Timely injection (e.g., 20% emamectin benzoate, 15–20 mL, higher dose) of diseased trees at this stage can also prevent disease progression to a certain extent, and some of these diseased plants may also recover. When the diseased pine trees have feeding wounds from longicorn beetles and the proportion of needle discolouration is greater than 50%, although the accuracy rate of initial diagnosis can reach 100%, at this stage, the diseased pine trees are basically outside the window for effective treatment. For the early stage of the diseased pine tree injection treatment, the tree can recover, and emamectin benzoate can remain in the tree for a certain period, continuing to play a role in disease prevention [22].

In this study, we found that pine trees with beetle feeding wounds may not necessarily be diseased, possibly because the nematode invasion of twigs may not occur every time during vector beetle supplementary nutrition, or there may not be sufficient numbers of nematodes to invade, and healthy pine trees are resistant to small numbers of nematode invasions [23,24]. When a small number of needles on the twigs begin to have a slight discolouration, the probability of diagnosis of PWD is greatly increased, and the implementation of agent treatment during this period is more effective.

There was a diseased pine tree with several branches with needles discoloured (middle stage II). After the trunk was injected with 20 mL of 20% emamectin benzoate, the external symptoms did not develop further, but the pine wood nematodes were still detected in the samples taken from its trunk, indicating that the agent entering the tree may not necessarily kill all nematodes in the tree. For live nematodes, the agent remaining in the tree can continue to play a role in inhibiting nematode growth and reproduction, but as time advances, the residue of the pesticide will gradually decrease until there is none [25]. At this time, the nematodes in the tree may continue to grow and reproduce, resulting in the death of the pine tree.

The content and residue of emamectin benzoate in trees is an important prerequisite for its effective application in the prevention and control of pine wood nematode disease at home and abroad [12,26,27]. However, the residue distribution dynamics of emamectin benzoate after trunk injection have not been systematically studied. In this study, 20% emamectin benzoate SL was injected during the growing season, and the side branches were obtained one year after the injection to detect the agent residues. The residual results of 20 mL and 10 mL of emamectin benzoate were 0.18 mg kg$^{-1}$ and 0.06 mg kg$^{-1}$, respectively, which was lower than that observed in previous research results [27,28]. This may be because most of the pine trees injected in this study were heavily branched, which dispersed the residues of agents in the tree, and the branches sampled in this experiment were the outermost fourth lateral branches, which led to low residues of agents. In addition,

when the drug is injected in the growing season, the secretion of rosin is vigorous, which may also have a certain impact on dredging. On the other hand, the disease did not develop further after the pine trees were injected in the early stage of the disease, indicating that the residual amount of the pesticide in the tree was enough to inhibit the propagation and expansion of pine wood nematodes, which could provide a reference for future research on the treatment of PWD with trunk injections.

## 5. Conclusions

Injection of 5–10mL of 20% emamectin benzoate SL at the early stage of infected pine trees can make pine trees recover to health, and injection of 20 mL (higher dose) of 20% emamectin benzoate SL at the middle stage of infected pine trees can significantly delay the death time of pine trees, with a certain probability of cure. The diagnostic accuracy of PWD based on the external symptoms of early stage II was high, and the treatment effect was good after injection at this stage, which can be used for rapid diagnosis and drug injection treatment in the pine forest.

**Author Contributions:** Conceptualization, experimental, data analysis, manuscript—writing, A.N.; sample collection, D.Y., H.C.; guarantor of the integrity of the entire study and approval of the final version of the manuscript, J.Y. All authors have read and agreed to the published version of the manuscript.

**Funding:** This research was supported by the National Key Research and Development Program of China (2021YFD1400903) and the "Early detection technology and products of pine wood nematode disease" Project of the National Forestry and Grassland Administration (ZD202001-01).

**Institutional Review Board Statement:** Not applicable.

**Informed Consent Statement:** Informed consent was obtained from all subjects involved in the study.

**Data Availability Statement:** The data presented in this study are available on request from the corresponding author.

**Conflicts of Interest:** The authors declare no conflict of interest.

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
