# Peer review of "Preliminary Study on Early Diagnosis and Rehabilitation Treatment of Pine Wood Nematode Disease Based on Partial Symptoms"

_forests, doi:10.3390/f14040657_

Round 1

Reviewer 1 Report

Dear authors, the work you have devoted to research of early diagnosis and rehabilitation of trees damaged with pine wilt disease is indeed very important and has high practical significance. The first question that occurred to me when reading the manuscript: What was the results of your group, basing on which you choose for treatment the insecticide emamectin benzoate SL against PWN? It would not be bad to point out that these are the results or to give a link to where this was published. Is the choice connected only with the successful transmission of the preparation in the tissues of the tree, its effective action on the nematode or on the insect carrier? It would be very interesting to see the mean diameter and height of the treated trees. I perfectly understand all the laboriousness of the work that you have carried out, but in order not to accuse you of speculation, the results must be supported by statistical levels of significance, then it would be possible to get away from the cumbersome description of the clinic of individual trees and show the repeatability of successful treatments.

Reviewer 2 Report

Minor changes.

The study is the development of a detailed method of treatment of the PWD infected pines (the protection of ancient and famous pine trees ) with a emamectin benzoate  SL preparation, combined with a preliminary visual diagnosis of the stages of pine wilt disease. It is concluded that it is possible to cure trees at the early stages of the PWD infection.

The process of the emamectin benzoate  SL penetration from the injection site into the tree branches as well as the retention time of the chemical inside the tree has been studied in detail. The concentration and two methods of nematicide injection have been accurately determined. For the first time, it has been proven that for an effective treatment, emamectin benzoate SL injection must be carried out during the growing season and sap movement (‘ the window for effective treatment’, page 7) rather than when the tree is dormant. This discovery, based on a new understanding of the nematode infestation process in the trunk and branches, it serves as the theoretical basis for the entire new methodology.

The authors attempt to use the statistical parameters into the values of results, but the samples (n) are small both for the experimental units and for the control, because the experiments were carried out using the naturally infected trees in the forest.

However, the statistical data in Table 5 for the concentration values of the emamectin benzoate residues in branches one year after the injection should be indicated (n, mean, s.d., CV)

Other notes:

P1-L8 Pinus

= in italics

P2-L74-75

=the name of the centrifuge (producer) and the rotation speed (g) and tube volume are to be indicated

nematode liquid

= suspension of nematodes, PWN suspension

Round 2

Reviewer 1 Report

Dear authors, you have partially taken into account my comments, but in order for me to give my consent to the publication, if you write word "SIGNIFICANT"I would like to see, fo example, the use of non-parometric statistical methods that would provide p-value , in addition, I propose to remove the categoricalness in the title, and indicating that these are the first or preliminary results of biotesting.
